## [Decision Letter · Decision Letter 0]

11 Sep 2022

PONE-D-22-13000Perceptions of adolescent mothers towards adolescent repeat childbirth in Soroti District, Teso sub-region, Uganda: A phenomenological studyPLOS ONE

Dear Dr. Mulalu,

Thank you for submitting your manuscript to PLOS ONE. After careful consideration, we feel that it has merit but does not fully meet PLOS ONE’s publication criteria as it currently stands. Therefore, we invite you to submit a revised version of the manuscript that addresses the points raised during the review process.

The reviewer report can be found at the end of this email, with detailed comments also provided in the annotated version of your manuscript in the attachment. Please read all comments carefully and fully address all concerns raised when you revise your manuscript. 

Furthermore, as this is a qualitative study, we recommend that you use the COREQ checklist, or other relevant checklists listed by the Equator Network, such as the SRQR, to ensure complete reporting (http://journals.plos.org/plosone/s/submission-guidelines#loc-qualitative-research). 

Please note that we have only been able to secure a single reviewer to assess your manuscript. We are issuing a decision on your manuscript at this point to prevent further delays in the evaluation of your manuscript. Please be aware that the editor who handles your revised manuscript might find it necessary to invite additional reviewers to assess this work once the revised manuscript is submitted. However, we will aim to proceed on the basis of this single review if possible. 

A marked-up copy of your manuscript that highlights changes made to the original version. You should upload this as a separate file labeled 'Revised Manuscript with Track Changes'.An unmarked version of your revised paper without tracked changes. You should upload this as a separate file labeled 'Manuscript'.

We look forward to receiving your revised manuscript.

Kind regards,

Debora Walker

Editorial Office

PLOS ONE

Journal Requirements:

Reviewers' comments:

Reviewer's Responses to Questions

**Comments to the Author**

1. Is the manuscript technically sound, and do the data support the conclusions?

Reviewer #1: Partly

2. Has the statistical analysis been performed appropriately and rigorously? 

Reviewer #1: N/A

3. Have the authors made all data underlying the findings in their manuscript fully available?

Reviewer #1: Yes

4. Is the manuscript presented in an intelligible fashion and written in standard English?

Reviewer #1: No

5. Review Comments to the Author

Reviewer #1: It was my pleasure to review this manuscript. However, there are areas that still require attention such as the choice for a phenomenological approach, data collection lack some important aspects, the demographic of study participants and summary of themes and sub themes are not clearly presented. When i read the participant quotes, it looks like this is a qualitative descriptive study as opposed to a phenomenological research. The area for further research need to be strengthened to state exactly what should be different to arrive at a different conclusion. I attached the document with track changes in it.

6. PLOS authors have the option to publish the peer review history of their article (what does this mean?). If published, this will include your full peer review and any attached files.

Reviewer #1: No

---

## [Author Response · Author response to Decision Letter 0]

14 Feb 2023

Thank you so much for sparing your valuable time to critically review this article. I can assure you that you have done a commendable job and I assure you that I have totally agreed to your decision. I have addressed all the comments as seen in the attachment letter entitled; response to reviewers. Thank you so much once again.

---

## [Editor Report · Decision Letter 1]

7 Mar 2023

Perceptions of adolescent mothers towards adolescent repeat childbirth in Soroti District, Teso sub-region, Uganda: A phenomenological study

PONE-D-22-13000R1

Dear Mr. Posiano Mulalu,

We’re pleased to inform you that your manuscript has been judged scientifically suitable for publication and will be formally accepted for publication once it meets all outstanding technical requirements.

Kind regards,

Daniel Optamutale Ashipala

Academic Editor

PLOS ONE

Additional Editor Comments (optional):

Thank you for the opportunity to review this manuscript. The manuscript has really improved, and I recommend it for publication. I sincerely appreciate the work.

Reviewers' comments:

I hope you are pleased.

---

## [Editor Report · Acceptance letter]

19 Apr 2023

PONE-D-22-13000R1 

Perceptions of adolescent mothers towards adolescent repeat childbirth in Soroti District, Teso sub-region, Uganda: A phenomenological study 

Dear Dr. Mulalu:

I'm pleased to inform you that your manuscript has been deemed suitable for publication in PLOS ONE. Congratulations! Your manuscript is now with our production department. 

Kind regards, 

on behalf of

Dr. Daniel Optamutale Ashipala 

Academic Editor

PLOS ONE